# Identification of Malignancies from Free-Text Histopathology Reports Using a Multi-Model Supervised Machine Learning Approach

**Victor Olago [1],\*, Mazvita Muchengeti [1,2], Elvira Singh [1,2] and Wenlong C. Chen [1,3]** 

[1]  National Cancer Registry, National Health Laboratory Service, 1 Modderfontein Road, Johannesburg 2003, South Africa; MazvitaM@nicd.ac.za (M.M.); ElviraS@nicd.ac.za (E.S.); WenlongC@nicd.ac.za (W.C.C.)

[2]  Division of Epidemiology and Biostatistics, School of Public Health, Faculty of Health Sciences, University of Witwatersrand, Johannesburg 2193, South Africa

[3]  Sydney Brenner Institute for Molecular Bioscience, Faculty of Health Sciences, University of the Witwatersrand, Johannesburg 2193, South Africa

\*  Correspondence: VictorO@nicd.ac.za

**Abstract:** We explored various Machine Learning (ML) models to evaluate how each model performs in the task of classifying histopathology reports. We trained, optimized, and performed classification with Stochastic Gradient Descent (SGD), Support Vector Machine (SVM), Random Forest (RF), K-Nearest Neighbor (KNN), Adaptive Boosting (AB), Decision Trees (DT), Gaussian Naïve Bayes (GNB), Logistic Regression (LR), and Dummy classifier. We started with 60,083 histopathology reports, which reduced to 60,069 after pre-processing. The F1-scores for SVM, SGD KNN, RF, DT, LR, AB, and GNB were 97%, 96%, 96%, 96%, 92%, 96%, 84%, and 88%, respectively, while the misclassification rates were 3.31%, 5.25%, 4.39%, 1.75%, 3.5%, 4.26%, 23.9%, and 19.94%, respectively. The approximate run times were 2 h, 20 min, 40 min, 8 h, 40 min, 10 min, 50 min, and 4 min, respectively. RF had the longest run time but the lowest misclassification rate on the labeled data. Our study demonstrated the possibility of applying ML techniques in the processing of free-text pathology reports for cancer registries for cancer incidence reporting in a Sub-Saharan Africa setting. This is an important consideration for the resource-constrained environments to leverage ML techniques to reduce workloads and improve the timeliness of reporting of cancer statistics.

**Keywords:** machine learning; multi-model supervised machine learning; text mining; text classification; natural language processing; cancer coding; flagging malignant reports

## 1. Introduction

The South African National Cancer Registry (NCR) is responsible for the registration of all malignancies, including histopathologically diagnosed malignancies, and annual reporting of cancer statistics for South Africa (SA) [1,2]. The NCR receives over 100,000 cancer pathology reports annually from pathology laboratories in SA [1,2]. All cancer pathology reports are coded according to the International Classification of Diseases for Oncology 3rd edition (ICD-O-3), reports are de-duplicated to identify index cancer cases, and the cancer statistics are calculated and reported annually [1,2]. The NCR database, since its inception in 1986, has over 1.2 million index cancer cases recorded [2].

The NCR receives pathology reports from both private and public laboratories throughout SA [1]. These reports are electronic and in free-text format. Trained data coders perform medical data abstraction and code the malignant reports using the ICD-O-3 topography and morphology classification for downstream analysis [3]. The medical data abstraction process is labor-intensive,

and the quality of the abstraction is prone to inter-person variability despite the training and experience of the data coder [3]. Ongoing quality assurance measures are deployed to ensure the NCR's reporting meets industry standards [4]; therefore, the reports are robust and reliable to inform cancer control policies and support robust research.

Despite technological improvements in the abstraction platform and workflows [5], two weaknesses persist. (1) First, there is the issue of inter-coder variability for medical abstraction. This process is labor-intensive [6] and prone to human errors. (2) Cancer reports from the reporting laboratories are identified using the pathologists/laboratory-reported Systematized Nomenclature of Medicine clinical terms (SNOMED-CT). Should pathologists/laboratories fail to report or incorrectly report the SNOMED-CT, then the pathology report may be missed by the NCR. This weakness is prominent in the laboratory reports submitted to the NCR by the public healthcare laboratories in SA. To address the workflow weaknesses and improve efficiency, the NCR must adopt text-mining techniques in the data processing workflow.

Text mining is the act of knowledge discovery in unstructured and structured textual data [7,8]. In the biomedical domain, text-mining techniques are used for process automation [7–10]. Such techniques can help to shorten data processing time, improve accuracy in text classification and abstractions, and reduce operation costs [6]. Text mining can be done by applying rule-based approaches, natural language processing, and/or machine learning (ML) [11].

The rule-based approaches include the use of regular expressions or creating a reference dictionary [12,13]. In this approach, a list of key terms is developed. Then, the model searches for the terms in the list and flags where the terms match [14,15]. For each row that matches the terms in the list, it assigns a positive or actual value. For the rows with no values in the list, it assigns a negative value or no outcome. This technique is ideal for classifying texts rows that are few or when the texts are always standard or structured [12]. In cases where the text is unstructured and the data keeps growing, the use of ML models is ideal [16,17].

ML models are broadly categorized into two groups: supervised or unsupervised [18]. Unsupervised ML models find patterns—for example, similarities and dissimilarities in the text or classes. Unsupervised models have no training data [19,20]. Supervised ML models use training data to perform classifications [21]. The training data act as the guide for performing the classification [21]. This makes that classification more accurate. Both supervised and unsupervised text-mining models are continuously being used in cancer registries and cancer research around the world [6,13,16,19,22–24].

Currently, the literature on text mining and text analytics work in the cancer registries across Sub-Saharan Africa is scarce. This work explores the use of text mining through multi-model supervised machine learning (MMSML) to identify histopathology reports describing malignancies among all histopathology reports in the public healthcare laboratories for the Western Cape Province in South Africa for the year 2016.

## 2. Methods

### 2.1. Ethical Considerations

This study is covered under the ethical clearance waiver granted to the NCR by the Human Research Ethics Committee of the University of the Witwatersrand.

### 2.2. Software and Hardware

We implemented this work entirely in Python 3.7 [25], running in Anaconda (Enterprise 4) [26] using Jupyter IPython Notebook (version 5.3.1) [27]. The Python modules pandas (version 0.24.2) [28], numpy (version 1.16.4) [29], regex (version 2.2.1) [30], nltk (version 3.4.4), and sklearn (version 0.21.2) [31] were used. We used a computer with an i7 processor and 32 gigabytes of RAM.

### 2.3. Data Source

All histopathology reports collated in the National Health Laboratory Service's (NHLS) Corporate Data Warehouse (CDW) [32] for the year 2016 in the Western Cape province were made available for this study. The NHLS is the central pathology laboratory service for the public healthcare sector in SA.

### 2.4. Pre-Processing

We assigned a unique row identifier (ID) for each record and then subset the dataset by retaining the three columns that contained the result text in free text format, the SNOMED-CT morphology codes, and the row ID. We plotted a word cloud on the result text to determine the word representation before data cleaning of the result text. We also generated a character count, word count, and unique word count before data cleaning. For each row in the result text, new lines, tabs, and extra spaces were replaced with a single space. Then, we picked from the start word patterns that are "*a*" to "*z*" in either lower or capital case, numbers 0 to 9, hyphens, apostrophes, and spaces. Then, we converted all the resulting text into lower case. We expanded the contracted words—for example "*don't*" to "*do not*"—and removed the words "no" and "not" from the stopwords list [15] because these are negation words in a sentence. We added the words "tel", "telephone", and "fax" to the stopwords list. Then, we removed all the words in the stopwords list from the resulting text.

A second word cloud was constructed after data cleaning to visualize the effect of the pre-processing. Character counts, word counts, and unique word counts were generated again after data cleaning. Comparison of the metadata data frame was done before and after pre-processing, and we excluded rows where the content was completely lost due to pre-processing. These rows, which initially had tabs and newlines values, were left with no result text after pre-processing (had null values).

### 2.5. Feature Engineering

This is the process of transforming input data to features that machine learning models can easily interpret to improve model performance [33,34]. This can be done by either reviewing the input features or letting the machine learning model select the most appropriate features [33,34]. We used Term Frequency-Inverse Document Frequency (TF-IDF) vectorizer with an ngram range of 1–3 to convert the raw text to the matrix of TF-IDF features. TF-IDF is a mathematical representation of a weight to a term or terms in a document [35]. It looks at how important a term or terms is/are with respect to the whole corpus or document [35]. The TF-IDF is given by the equation below:

$$W(d,t) = TF(d,t) * log\left(\frac{N}{df(t)}\right)$$

where *d* is document, *t* is term, *df* is document frequency, *TF* is term frequency, and *N* is the total number.

Through feature engineering, we were able to drop words that the TF-IDF vectorizer gave more weight to when paired with other words that do not contribute to the actual classification goal. These included words such as comment, diagnosis, final diagnosis, immunohistochemistry, microscopic examination, etc.; these words were subheadings in the reports. We sampled some of the most important words, bigrams, and trigrams for this classification and attached a list in the Supplementary Materials.

Then, we fitted the features to the encoded value labels. Since TF-IDF generates many features, it is impossible to use all the features to perform classification. Therefore, we used a dimensionality reduction feature called Truncated Singular Value Decomposition (SVD) [36] through a topic modeling technique called Latent Semantic Analysis [31].

### 2.6. Classification

We sampled records with SNOMED-CT morphology codes to create classes of malignancy status for the training data. The cancer morphology codes are five-digit codes ranging from 8000/0 to

9992/9. The first four digits indicate the specific histologic term [37], while the code after the backslash represents the behavior code. The behavior code can be as follows: 0 is benign, 1 is uncertain whether malignant or benign, 2 is carcinoma in situ, 3 is the malignant primary site, 6 is the malignant metastatic site, and 9 is malignant, uncertain whether primary or metastatic site [37].

Using regular expressions [30], we were able to construct classes for "Malignant", "Non-malignant", and "No diagnosis". We performed the MMSML classification in scikit-learn in Python [31]. We randomly sampled 5000 rows each from "Malignant" and "Non-malignant" classes and 1000 rows from "No diagnosis" to create the training data. Then, we used the Label Encoder to turn the labels into numbers that are 0, 1, and 2 for "Non-malignant", "Malignant", and "No diagnosis", respectively. We applied a multiclass classification approach. We split the training data into "X" and "y" where "X" is the result text that also contains the features for the classification and "y" is the encoded labels.

We evaluated the model performance by running non-optimized classification algorithms. We used the train test split by stratification method in scikit-learn [31]. The test size was "0.3", the random state was "three", stratification was "yes", and shuffle was set to "true". Then, we optimized the models by performing hyperparameter tuning in GridSearchCV. The scikit-learn library allowed us to stack these models together, thereby making it possible to compare the performance of each algorithm [31]. The algorithms used in this model are briefly explained below.

### 2.6.1. Gaussian Naïve Bayes (GNB)

GNB is a classifier is based on Bayes theorem [35]. It relies on the conditional probability to predict the outcome of an occurrence [35]. For example, if documents $n$ fit into $k$ categories where $k \in \{c_1, c_2, \ldots, c_k\}$, then the predicted output is $c \in C$. The model function is given as below:

$$P(c|d) = \frac{P(d|c)P(c)}{P(d)}$$

where $d$ is documents and $c$ indicates classes.

### 2.6.2. Adaptive Boosting (AB)

AB was discovered by Freund and Schapire; this algorithm works by reweighing the examples in the training set to improve the classification accuracy [38]. It converts any algorithm with an accuracy higher than guessing to a higher performance [38]. A boost classifier is giver by the function below:

$$F_T(x) = \sum_{t=1}^{T} f_t(x)$$

where $f_t$ is a weak learner that takes an object $x$ as input and return the class it belongs to.

### 2.6.3. Logistic Regression (LR)

LR uses a logistic function to predict a given outcome [39]. LR is also referred to as the maximum entropy model in the multiclass text classification domain [39]. To perform multiclass text classification, LR must be regularized. This is possible by adding a regularization term $w^T w/2$, and a regularized logistic regression is given by the function below;

$$\min_{w} f(w) \equiv \frac{1}{2} w^T w + C \sum_{i=1}^{l} log\left(1 + e^{-y_i w^T x_i}\right)$$

where $C > 0$ is a parameter set by users. The function estimates ($w$) weight by (*min*), minimizing the negative log-likelihood.

### 2.6.4. Stochastic Gradient Descent (SGD)

SGD is mostly used in large-scale machine learning problems since the computational complexity of machine learning becomes a limiting factor in very large datasets [40]. SGD addresses the complexity by having a faster convergence [41], as the SGD algorithm learns by randomly obtaining examples from the ground truth without necessarily taking into consideration the previous iterations [40]. Every iteration in SGD updates the weights based on the gradient from the randomly picked example [40]:

$$w_{t+1} = w_t - \gamma_t \nabla_w Q(z_t, w_t)$$

where $z_t$ is the random example picked, $w_t$, $t = 1, \ldots, t = n$, is the stochastic process that depends on the randomly picked example [40]. SGD is derived from Batch Gradient Descent (BGD) [41]. BGD is meant for small datasets, while SGD works well in large datasets. We used a constant learning rate to maintain class labels.

### 2.6.5. K-Nearest Neighbor (KNN)

This is a non-parametric algorithm, which considers the closest neighbor to the point of prediction [35]. For example, consider a document with the $x$ training set; the algorithm will find all the k neighbors of $x$. Since there may be lots of overlap in the neighbors, the algorithm assigns a score to the $k$ neighbors and only puts the $k$ with the highest scores depending on the value of $x$. We used weight-adjusted KNN that uses the TF-IDF weight vectors for the classification [35], where the KNN weighted cosine measure was derived as follows:

$$\cos(x, y, w) = \frac{\sum_{t \in T}(x_t \times w_t) \times (y_t \times w_t)}{\sqrt{\sum_{t \in T}(x_t \times w_t)^2} \times \sqrt{\sum_{t \in T}(y_t \times w_t)^2}}$$

where $T$ is the set of words, and $x_t$ and $y_t$ are the term frequencies. The training set ($d \in D$), where $N_d = \{n_1, n_2, \ldots, n_k\}$ is the set of k-nearest neighbors of $d$. The similarity sum of $d$ neighbors that belongs to class c given by $N_d$ defined as:

$$S_c = \sum_{n_i \in N; C(n_i) = c} \cos(d, n_i, w).$$

The similarity total is given as below:

$$T = \sum_{c \in C} S_c$$

and $d$ contribution defined in the terms of $S_c$ of classes $c$.

### 2.6.6. Support Vector Machine (SVM)

Originally developed as a binary classifier, but with the recent advancement in technology, SVM algorithms have improved to non-binary and multiclass classifications models [35]. SVM uses either linear or non-linear kernels to perform classification [35]. We used multiclass SVM by applying one versus the rest while generating classification features from TF-IDF [35]. To get proper classification, we used a string kernel [35]. The string kernel uses $\Phi(.)$ to map the string in the feature space. By using the spectrum kernel, which counts the number of times a word appears in string $x_i$ as a feature map where defining feature maps from $x \to R^{lk}$:

$$\Phi(x) = \Phi_j(x)_{j \in \sum^k}$$

where the kernel $\Phi_j(x) =$ number of $j$ feature appears in $x$.

The feature map $\Phi_i(x)$ is then generated by sequence $x_i$ and kernel defined as follows:

$$F = \sum^k$$

$$K_i(x, x') = \langle \Phi_i(x), \Phi_i(x') \rangle$$

### 2.6.7. Decision Trees (DT)

This classifier performs classification by creating a tree based on attributes of data points [35,42]. Classification is performed by getting attributes with the largest information gain as the parent's node, then using cross-entropy to evaluate the performance of the classification [42]. For example, consider an attribute $A$ with $k$ distinct value divides the training set $E$ in subsets of $\{E_1, E_2, \ldots, E_k\}$.

### 2.6.8. Random Forest (RF)

This is an ensemble learning method for text classification; it works by generating random decision trees [35]. It is faster to train for faster classification, though it is quite slow to make predictions [35]. The algorithm has been improved to have convergence as margin measures ($mg(X,Y)$) with indicator function $I(.)$ as below:

$$mg(X, Y) = av_k I(h_k(X) = Y) - \max_{j \neq Y} av_k I(h_k(X) = j).$$

The predictions in RF are assigned based on voting as follows:

$$\delta_V = arg \max_i \sum_{j:j \neq j} I_{\{r_{ij} > r_{ij}\}}$$

such that $r_{ij} + r_{ij} = 1$.

### 2.7. Model Optimization

Hyper-tuning of the model parameters was done, and the classification was run through a GridSearchCV to improve the model performance [31]. GridSearchCV implements a fit and score method [31]. We performed 5-fold cross-validation on a GridSearchCV, and the best model was selected based on the score each fold returned. This allowed us to optimize all the algorithms in the model except for the dummy classifier. Table 1 shows the optimized parameters that we used to perform classification.

**Table 1.** Classification parameters for the optimized models. SGD: Stochastic Gradient Descent, RF: Random Forest, SVM: Support Vector Machine, KNN: K-Nearest Neighbor, AB: Adaptive Boosting, DT: Decision Tree, LR: Logistic Regression, GNB: Gaussian Naïve Bayes.

| Model | Parameters |
|---|---|
| SGD | Alpha = 0.0001, average = False, class_weight = None, early_stopping = False, epsilon = 0.1, eta0 = 0.0, fit_intercept = True, l1_ratio = 0.15, learning_rate = 'optimal', loss = 'log', max_iter = 1000, n_iter_no_change = 5, n_jobs = None, penalty = 'l2', power_t = 0.5, random_state = None, shuffle = True, tol = 0.001, validation_fraction = 0.1, verbose = 0, warm_start = False |
| RF | bootstrap = True, ccp_alpha = 0.0, class_weight = None, criterion = 'gini', max_depth = None, max_features = 'auto', max_leaf_nodes = None, max_samples = None, min_impurity_decrease = 0.0, min_impurity_split = None, min_samples_leaf = 1, min_samples_split = 2, min_weight_fraction_leaf = 0.0, n_estimators = 100, n_jobs = None, oob_score = False, random_state = None, verbose = 0, warm_start = False |
| SVM | C = 1.0, break_ties = False, cache_size = 200, class_weight = None, coef0 = 0.0, decision_function_shape = 'ovr', degree = 3, gamma = 'scale', kernel = 'rbf', max_iter = −1, probability = True, random_state = None, shrinking = True, tol = 0.001, verbose = False |
| KNN | algorithm = 'auto', leaf_size = 30, metric = 'minkowski', metric_params = None, n_jobs = None, n_neighbors = 5, p = 2, weights = 'uniform' |

**Table 1.** *Cont.*

| Model | Parameters |
|---|---|
| AB | algorithm = 'SAMME.R', base_estimator = None, learning_rate = 1.0, n_estimators = 50, random_state = None |
| DT | ccp_alpha = 0.0, class_weight = None, criterion = 'gini', max_depth = None, max_features = None, max_leaf_nodes = None, min_impurity_decrease = 0.0, min_impurity_split = None, min_samples_leaf = 1, min_samples_split = 2, min_weight_fraction_leaf = 0.0, presort = 'deprecated', random_state = None, splitter = 'best' |
| LR | C = 1.0, class_weight = None, dual = False, fit_intercept = True, intercept_scaling = 1, l1_ratio = None, max_iter = 100, multi_class = 'auto', n_jobs = None, penalty = 'l2', random_state = 8, solver = 'lbfgs', tol = 0.0001, verbose = 0, warm_start = False |
| GNB | priors = None, var_smoothing = $1 \times 10^{-9}$ |

## 2.8. Evaluation

We evaluated our models by calculating the accuracy, precision, recall, $F_1$-score, misclassification rate (error rate), micro-average, and macro-average. We achieved this by plotting a Confusion Matrix (CM), Receiver Operating Characteristics (ROC), and Area Under Curve (AUC) for various algorithms [35]. CM is a table used to measure the performance of a classification model by getting counts of predicted values against actual values [35]. ROC and AUC measure the performance of the classification at various threshold settings [35]. ROC is the probability curve, while AUC is a measure of separation of the classes [35]. The plotting of confusion matrices, ROC, and AUC are possible from calculating elements such as the True Positive (TP), False Positive (FP), True Negative (TN), False Negative (FN), True Positive Rate (TPR), and False Negative Rate (FNR) [35].

The example reports in Table 2 show examples of the reports before and after cleaning and their respective SNOMED-CT classes.

**Table 2.** Example of the histopathology reports for the three classes.

| Report before Pre-Processing | Report after Pre-Processing | Text Class | Class |
|---|---|---|---|
| EPISODE NUMBER: \nXX1111111\n\n\nCLINICAL HISTORY: \nMULTIPLE GROWTH ABDOMEN RIGHT THIGH MOBILE 2 X 2CM. NIL SKIN CHANGE.\nSUBCUTANEOUS LESION RIGHT THIGH. LIPOMA.\n\n\nMACROSCOPY: \nSPECIMEN LABELED ¿LIPOMA RIGHT THIGH¿ CONSISTS OF A YELLOW FRAGMENT OF TISSUE MEASURING 15 X 8 X 5MM.\n\n\nMICROSCOPY:\nSECTION SHOWS LOBULES OF MATURE ADIPOCYTES WITH NUMEROUS INTERVENING SMALL BLOOD VESSELS. FIBRIN THROMBI ARE CONSPICUOUS WITHIN THE VESSELS. THERE IS NO EVIDENCE OF ATYPIA OR MALIGNANCY.\n\nPATHOLOGICAL DIAGNOSIS:\nSUBCUTANEOUS TISSUE OF RIGHT THIGH, EXCISIONAL BIOPSY: ANGIOLIPOMA.\n\n\nREPORTED BY: DR YYYYYY\n | episode number: xx1111111 clinical history: multiple growth abdomen right thigh mobile 2 × 2 cm nil skin change subcutaneous lesion right thigh lipoma macroscopy: specimen labeled lipoma right thigh consists yellow fragment tissue measuring 15 × 8 × 5 mm microscopy: section shows lobules mature adipocytes numerous intervening small blood vessels fibrin thrombi conspicuous within vessels no evidence atypia malignancy pathological diagnosis: subcutaneous tissue right thigh excisional biopsy: angiolipoma reported by: dr yyyyyy | Non-malignant | 0 |
| EPISODE NUMBER: \nXX2222222\n\n\nCLINICAL HISTORY: \nSKIN CANCER LEFT LOWER LEG. LONG STITCH = LATERAL, SHORT STITCH = SUPERIOR.\n\n\nMACROSCOPY:\nSPECIMEN LABELED ¿BCC LEFT LEG¿ CONSISTS OF AN ELLIPSE OF SKIN WITH A LARGE POLYPOID ULCERATED TUMOR MEASURING 65MM IN THE LONG AXIS AND 45MM IN THE SHORT AXIS. THE ULCER MEASURES 35 X 30MM.\n\n\nMICROSCOPY: \nSECTIONS SHOW A MODERATELY DIFFERENTIATED KERATINIZING SQUAMOUS CARCINOMA.\nMARGINS: \nLATERAL SKIN MARGIN 4MM\nMEDIAL 6MM\nPROXIMAL 12MM\nDISTAL 10MM\nDEEP RESECTION MARGIN 4-5MM\n\n\nPATHOLOGICAL DIAGNOSIS:\nSKIN OF LOWER LEG, EXCISION: SQUAMOUS CARCINOMA\n\n\nREPORTED BY: PROF XXXXXX\n | episode number: xx2222222 clinical history: skin cancer left lower leg long stitch lateral short stitch superior macroscopy: specimen labeled bcc left leg consists ellipse skin large polypoid ulcerated tumor measuring 65 mm long axis 45 mm short axis ulcer measures 35 × 30 mm microscopy: sections show moderately differentiated keratinizing squamous carcinoma margins: lateral skin margin 4 mm medial 6 mm proximal 12 mm distal 10 mm deep resection margin 4–5 mm pathological diagnosis: skin lower leg excision: squamous carcinoma reported: prof xxxxxx | Malignant | 1 |
| EPISODE NUMBER: \nXX3333333\n\n\nCLINICAL HISTORY: \nBMT\n\n\nPATHOLOGICAL DIAGNOSIS:\nIMMUNOHISTOCHEMISTRY PERFORMED AT THE DEPARTMENT OF ANATOMICAL PATHOLOGY, GROOTE SCHUUR HOSPITAL.\n\nRETURNED TO REFERRING CENTRE FOR REPORTING.\n | episode number: xx3333333 clinical history: bmt pathological diagnosis: immunohistochemistry performed department anatomical pathology groote schuur hospital returned referring centre reporting | No Diagnosis | 2 |

## 3. Results

### 3.1. Pre-Processing

A total of 60,083 histology reports were registered by the NHLS for the Western Cape Province in 2016. The mean character count before pre-processing was 1032.12 with a standard deviation of 832.17.

The character count of the reports before pre-processing ranged from 1 to 16,961, which changed to 0 to 12,419 after pre-processing. The word count of the reports before pre-processing ranged from 1 to 5613, which changed from 0 to 1607 after pre-processing. Table 3 shows the summary statistics of the text results before and after pre-processing.

**Table 3.** The summary statistics of the histopathology reports before and after pre-processing.

| | Character Count before Pre-Processing | Character Count after Pre-Processing | Word Count before Pre-Processing | Word Count after Pre-Processing | Unique Words before Pre-Processing | Unique Words after Pre-Processing | Percentage Change |
|---|---|---|---|---|---|---|---|
| **Row count** | 60,083 | 60,068 | 60,083 | 60,068 | 60,083 | 60,068 | |
| **mean** | 1032.12 | 863.09 | 143.32 | 110.12 | 88.10 | 81.69 | 14.41 |
| **Standard deviation** | 832.17 | 667.63 | 154.34 | 87.82 | 54.05 | 47.76 | 15.79 |
| **min** | 1 | 0 | 1 | 0 | 1 | 0 | −125 |
| **25%** | 626 | 538 | 74 | 67 | 60 | 57 | 6.67 |
| **50%** | 836 | 702 | 105 | 89 | 79 | 74 | 13.79 |
| **75%** | 1180 | 982 | 164 | 125 | 104 | 96 | 20 |
| **max** | 16,961 | 12,419 | 5613 | 1607 | 983 | 884 | 90.14 |

We plotted the distribution characteristics of the characters, word count, and unique word count before and after pre-processing of the histopathology reports (Figure 1). The characters count, word count, and unique word count remained fairly unchanged from before to after pre-processing, as portrayed in the shapes of the distribution curves (Figure 1).

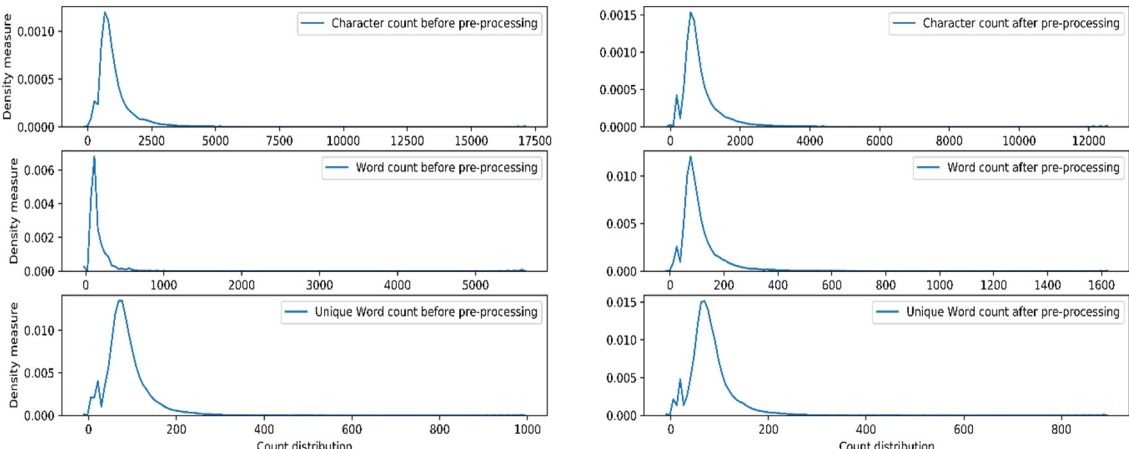

**Figure 1.** The distribution of character, word count, and unique word count of the histopathology reports before and after pre-processing.

We plotted two word clouds (Figure 2). Word clouds were used to visualize the keywords in the histopathology reports [43]. From the two word clouds, the effects of pre-processing were highlighted: the text changed to lower case, the new line tags were eliminated but generally, while the content of the histopathology reports remained unchanged.

The average word percentage change for before and after pre-processing was 14.41% (Figure 3). An expansion of shortened words led to a stopwords removal and a decreasing of words for some histopathology reports while other reports remained unchanged.

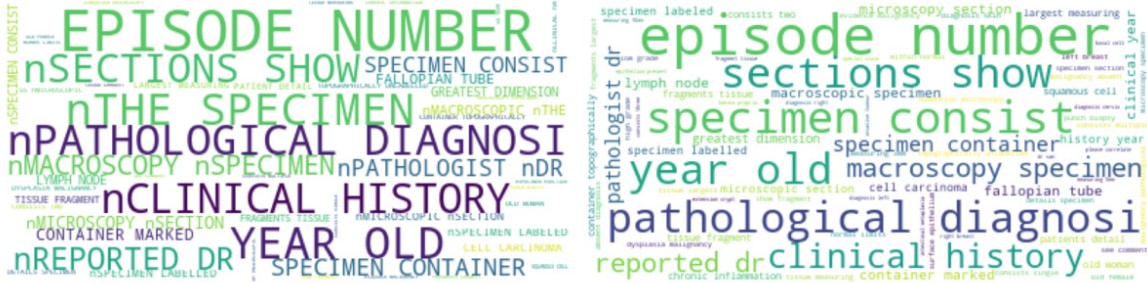

**Figure 2.** Word cloud for before and after pre-processing of the histopathology reports.

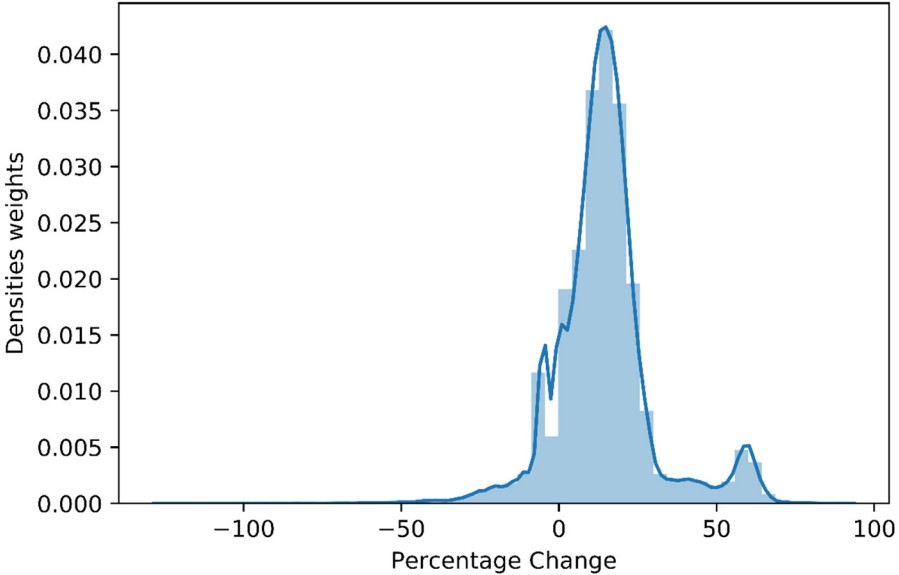

**Figure 3.** Words percentage change after pre-processing.

### 3.2. Classification

We randomly sampled 11,000 reports (from the total of 60,068 that survived pre-processing), where the SNOMED-CT codes indicated the "Malignant", "Non-malignant", and "No diagnosis" classifications for our training set. Then, we split the training dataset into two where 70% was for the training set and 30% was for the test set. We performed classification without optimization (Table 4).

**Table 4.** Models performance before optimization.

| Model | Accuracy (%) | Precision (%) | Recall (%) | $F_1$-Score (%) |
|-------|-------------|---------------|------------|------------------|
| SVM | 96 | 97 | 97 | 97 |
| LR | 95 | 96 | 96 | 96 |
| KNN | 95 | 96 | 96 | 96 |
| SGD | 95 | 96 | 96 | 96 |
| RF | 95 | 96 | 96 | 96 |
| DT | 90 | 93 | 92 | 92 |
| GNB | 85 | 89 | 88 | 88 |
| AB | 79 | 85 | 82 | 84 |
| Dummy | 41 | 32 | 32 | 32 |

We plotted the CM, ROC, and AUC to show the classification rates between various classes and the average classification as shown in Figures 4–7.

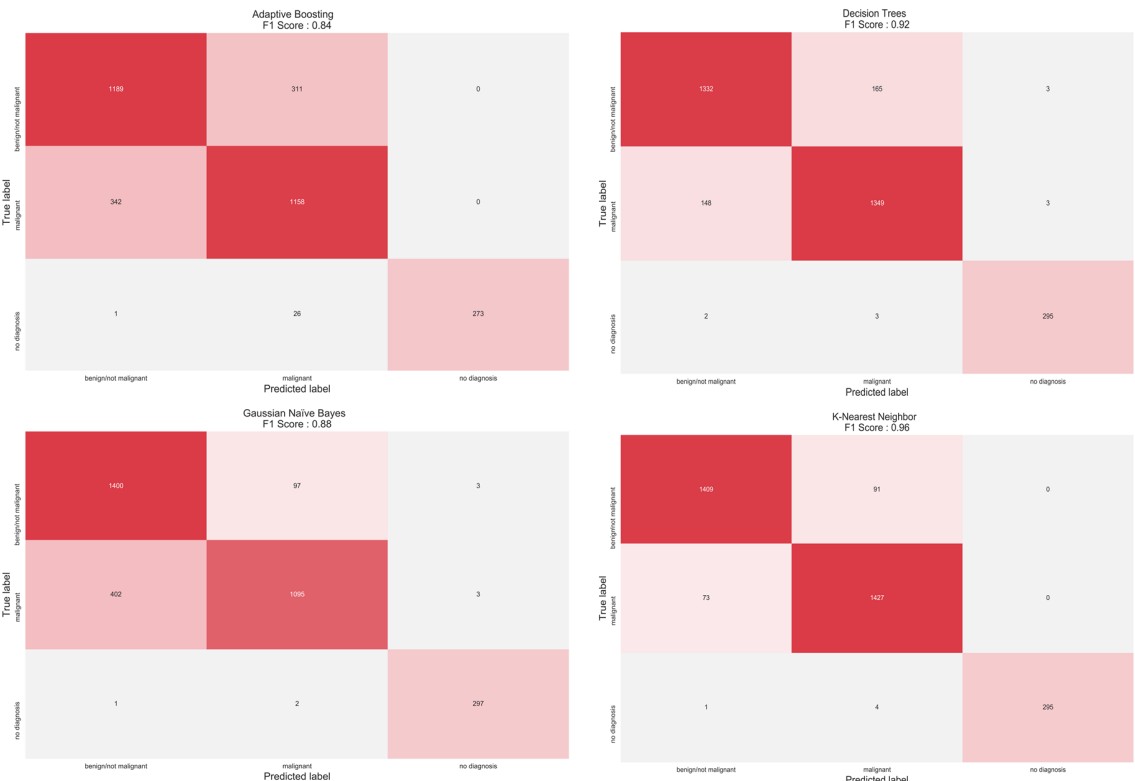

**Figure 4.** Performance evaluation using the confusion matrix of AB, DT, GNB, and KNN.

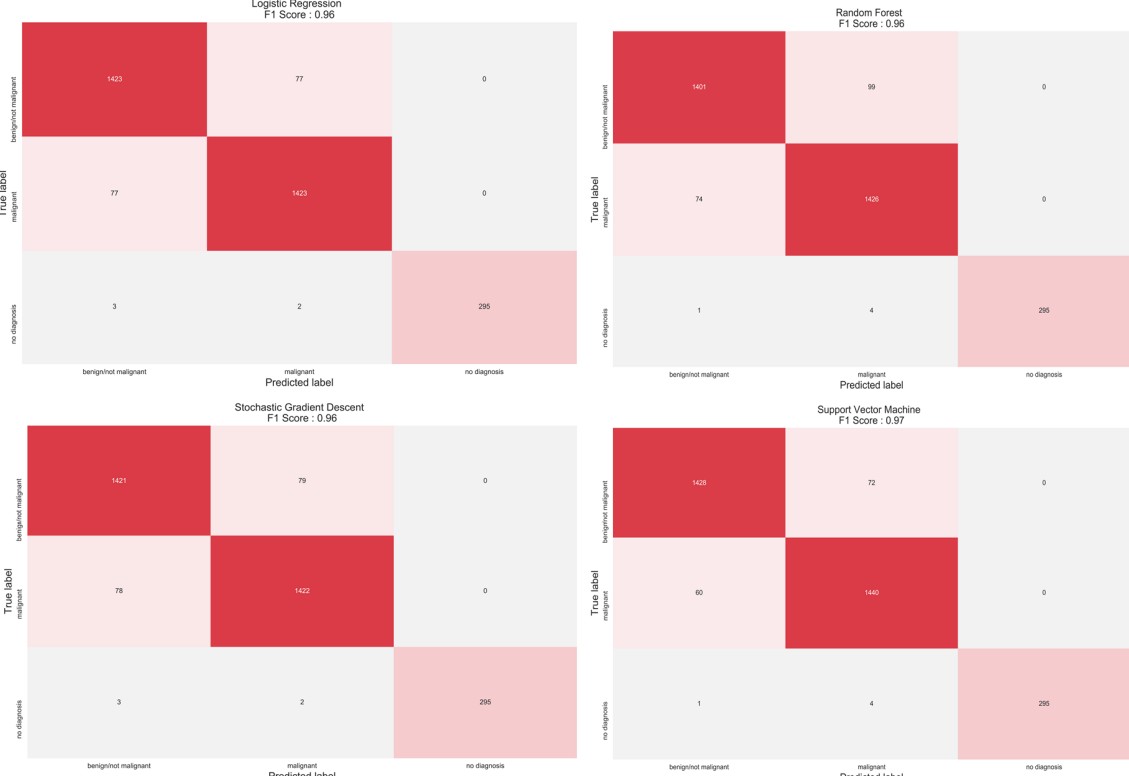

**Figure 5.** Performance evaluation using the confusion matrix of LR, RF, SGD, and SVM.

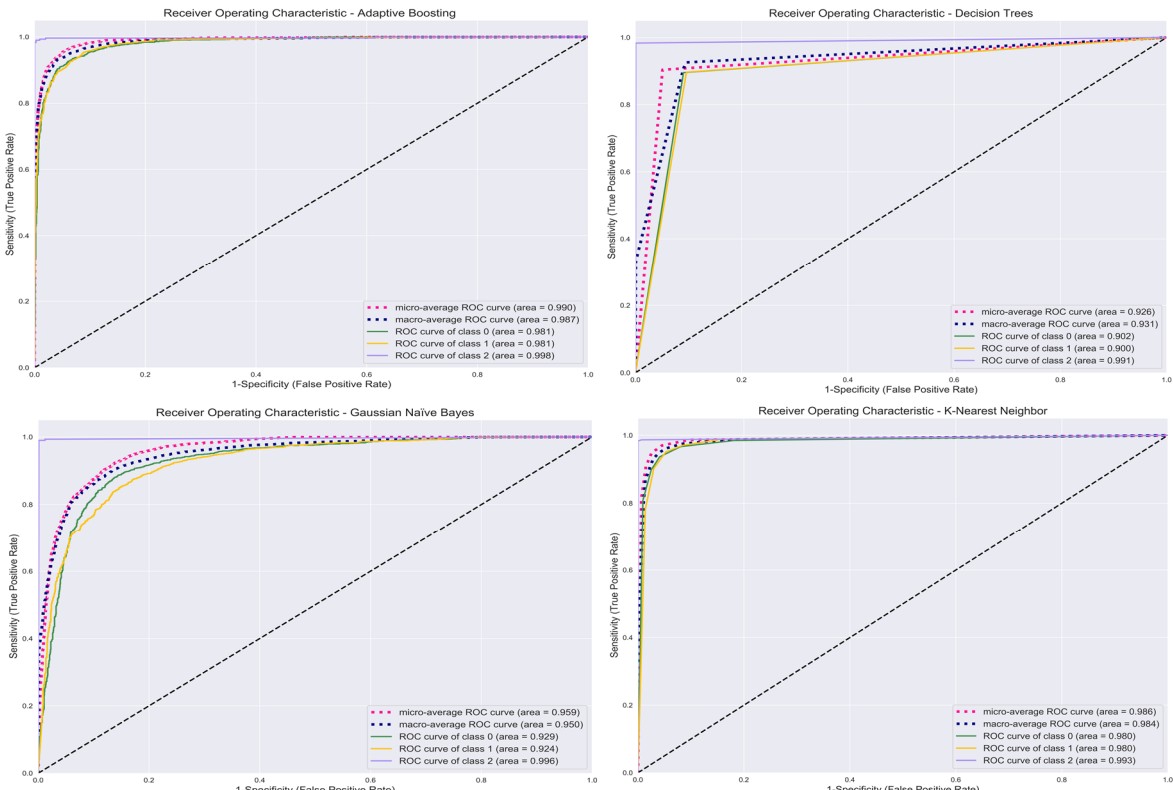

**Figure 6.** Performance evaluation using ROC for AB, DT, GNB, and KNN.

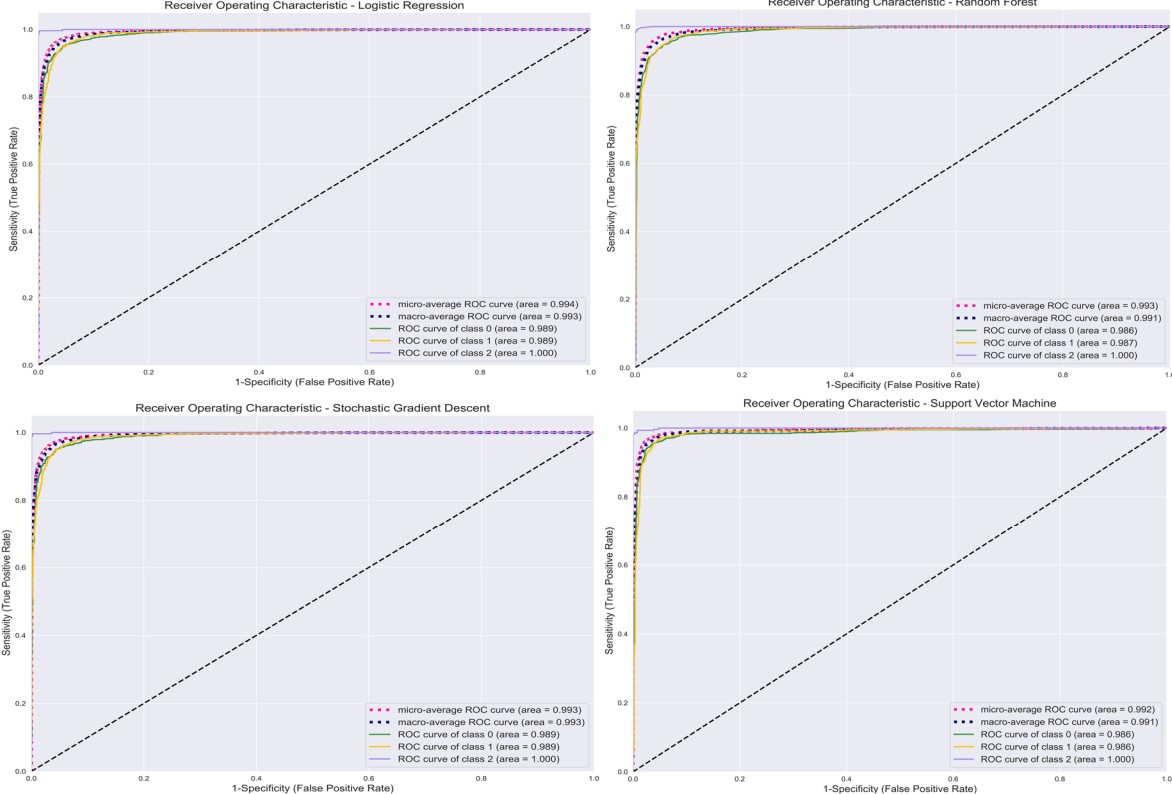

**Figure 7.** Performance evaluation using ROC for LR, RF, SGD, and SVM.

After model optimization, we performed the actual classification, as shown in Table 5. The table presents the cross-tabulation between the labels before and the outcome of the predictions from the

model. Then, we calculated the error rates for each algorithm in the model; the error rates were calculated using all reports with labels. The error rates and the run time for each algorithm in the model are presented in Table 6.

**Table 5.** Actual classification.

| Model | Predicted | Label before | | | | Total |
|---|---|---|---|---|---|---|
| | | **Non-Malignant** | **Malignant** | **No Diagnosis** | **No Label** | |
| SDG | Non-malignant | 5193 | 689 | 12 | 36,019 | |
| | Malignant | 208 | 10,000 | 3 | 5831 | 16,042 |
| | No diagnosis | 0 | 1 | 1283 | 830 | 2114 |
| | Total | 5401 | 10,690 | 1298 | 42,680 | |
| SVM | Non-malignant | 5192 | 351 | 8 | 32,662 | 38,213 |
| | Malignant | 209 | 10,338 | 7 | 9188 | 19,742 |
| | No diagnosis | 0 | 1 | 1283 | 830 | 2114 |
| | Total | 5401 | 10,690 | 1298 | 42,680 | |
| RF | Non-malignant | 5379 | 282 | 1 | 29,199 | 34,861 |
| | Malignant | 22 | 10,408 | 0 | 12,543 | 22,973 |
| | No diagnosis | 0 | 0 | 1297 | 938 | 2235 |
| | Total | 5401 | 10,690 | 1298 | 42,680 | |
| KNN | Non-malignant | 5188 | 536 | 6 | 26,419 | 32,149 |
| | Malignant | 213 | 10,153 | 8 | 15,425 | 25,799 |
| | No diagnosis | 0 | 1 | 1284 | 836 | 2121 |
| | Total | 5401 | 10,690 | 1298 | 42,680 | |
| DT | Non-malignant | 5359 | 559 | 1 | 26,937 | 32,856 |
| | Malignant | 40 | 10,126 | 1 | 13,900 | 24,067 |
| | No diagnosis | 2 | 5 | 1296 | 1843 | 3146 |
| | Total | 5401 | 10,690 | 1298 | 42,680 | |
| AB | Non-malignant | 4142 | 2876 | 0 | 25,881 | 32,899 |
| | Malignant | 1259 | 7810 | 17 | 15,834 | 24,920 |
| | No diagnosis | 0 | 4 | 1281 | 965 | 2250 |
| | Total | 5401 | 10,690 | 1298 | 42,680 | |
| GNB | Non-malignant | 5000 | 3044 | 7 | 31,308 | 39,359 |
| | Malignant | 398 | 7634 | 3 | 10,258 | 18,293 |
| | No diagnosis | 3 | 12 | 1288 | 1114 | 2417 |
| | Total | 5401 | 10,690 | 1298 | 42,680 | |
| LR | Non-malignant | 5148 | 471 | 12 | 33,828 | 39,459 |
| | Malignant | 253 | 10,218 | 3 | 8022 | 18,496 |
| | No diagnosis | 0 | 1 | 1283 | 830 | 2114 |
| | Total | 5401 | 10,690 | 1298 | 42,680 | |

**Table 6.** Misclassification error rates and model run times.

| Model | Misclassification Rates (%) | Approx. (Run Time) |
| --- | --- | --- |
| SGD | 5.25 | 20 min |
| SVM | 3.31 | 2 h |
| RF | 1.75 | 8 h |
| KNN | 4.39 | 40 min |
| DT | 3.50 | 2 h |
| AB | 23.90 | 50 min |
| GNB | 19.94 | 4 min |
| LR | 4.26 | 10 min |

## 4. Discussion

This study demonstrates the possibilities of integrating ML models to process cancer reports. Data labels make it possible to assign the SNOMED-CT codes to the unlabeled data. This is very important, as in our data, 8.17% of the pathology reports are not assigned any SNOMED-CT codes, while 10% of the assigned SNOMED-CT codes are misclassified. Considering the increasing cancer burden in low and middle-income countries (LMIC) worldwide due to changes in the lifestyle and environmental factors [5,44], more cancer reports are being collected in cancer registries. This requires faster and more efficient means of data processing to meet the demand for the timely and accurate reporting of cancer statistics.

By integrating ML models in data processing, it is possible to achieve timely data processing for the increased reporting load. For example, NCR collects more than 100,000 raw reports per annum for reporting purposes [1]. The number of raw reports is expected to increase with the increase in population size and the rise of cancers cases in LMICs.

Using ML techniques such as TF-IDF to generate classification features per classes assigned, it is possible to classify records without creating a reference/word dictionary. This also makes it possible to classify records with variability, since the majority of pathology records do not follow definite standard reporting guidelines, and variation exists amongst pathologists. Despite TF-IDF generating many classification features, a dimensional reduction of features using SVD makes it possible to reduce the number of classification features, thereby reducing the classification time while increasing the accuracy. This allows the appropriate features to be assigned to the appropriate classes for the training data promptly. This is evident, as we first tested the models by splitting the training dataset into two and using 30% for testing each classification algorithm.

RF performs well with least misclassification rates followed by SVM, DT, LR, and then KNN; this is also mirrored in the $F_1$-score of the five algorithms except for DT. The $F_1$-score for DT during training and optimization were at 92%, but there was improvement during the actual classification (Table 5, Figures 4 and 6). The classification rates of RF, SVM, LR, KNN, and SGD for each class were at 97% and above for the five algorithms, while the micro- and macro-classification were at 98% and above, as shown the ROC curves (Figures 6 and 7). Even though AB and GNB algorithms takes a short time to train, optimize, and perform classification with, they have high misclassification rates and are not appropriate to perform the classification of histopathology reports. The run time for RF is a limitation, but it had the least number of misclassification rates on the label data, and therefore, this showed its classification strength. The model is also known to train faster, but it takes longer to perform optimization [35]. LR can still be applied in text classification tasks for histopathology reports since it had a misclassification error of below 5% and an $F_1$-score of 96%. LR also takes a short time to train, optimize, and perform classification.

When we explored the misclassified reports in the model, all the algorithms uniformly misclassified 59 reports: 2 reports were predicted as malignant but were non-malignant, while 57 reports were predicted as non-malignant but were malignant. Our study did not incorporate Deep Learning models that are gaining popularity in the Natural Language Processing domain [35]. It would be ideal to try

such models in the histopathology reports and measure their performance. We were also not able to incorporate Multinomial Naïve Bayes (MNB) classifier, which is an improved GNB and has an enhanced performance compared to GNB [35]. The SVD dimensionality reduction applied in this study generates classification features with negative values, which made the MNB algorithm generate value error.

Performing classification with ML models saves more time compared to human coding [6] and is more accurate compared to rule-based approaches [13] in cases where datasets are big and have no standard structure. This helps to cope with constantly increasing heterogeneous data when such models are incorporated in workflow pipelines. This is a major strength, as there are no or little adjustments made to the model compared to rule-based approaches [45].

## 5. Conclusions

Our study demonstrated the possibility of applying ML techniques in the processing of free-text pathology reports for cancer registries for cancer incidence reporting in a Sub-Saharan African setting. This is an important consideration for resource-constrained environments to leverage ML techniques to reduce workloads and increase productivity. We can apply ML models to improve data processing efficiency and report misclassification.

RF, though it takes a long time to train and optimize, has the least misclassification rates and therefore would be recommended for performing the classification of histopathology reports. DT had the third least misclassification rates after SVM, which makes RF, SVM, and DT classifiers appropriate for text classification of the histopathology reports.

**Supplementary Materials:** The following are available online at http://www.mdpi.com/2078-2489/11/9/455/s1.

**Author Contributions:** Conceptualization, W.C.C. and V.O.; methodology, V.O.; software, V.O.; writing—original draft preparation, V.O.; writing—review and editing, M.M., E.S.; supervision, E.S. All authors have read and agreed to the published version of the manuscript.

**Funding:** This research received no external funding.

**Acknowledgments:** The authors would like to thank Trevor Bell for his input and contributions to this study.

**Conflicts of Interest:** The authors declare no conflict of interest.

## Abbreviations

The following abbreviations are used in this manuscript:

| | |
|---|---|
| SNOMED CT | Systematized Nomenclature of Medicine—Clinical Terms |
| ICD-O | International Classification of Diseases for Oncology |
| GridSearchCV | GridSearch Cross-Validation |
| SSA | Sub-Saharan Africa |
| SA | South Africa |
| NCR | South African National Cancer Registry |
| NHLS | National Health Laboratory Service |
| ML | Machine Learning |
| ID | Identification |
| SGD | Stochastic Gradient Descent |
| SVM | Support Vector Machine |
| AB | Adaptive Boosting |
| GNB | Gaussian Naïve Bayes |
| LR | Logistic Regression |
| DT | Decision Trees |
| KNN | K-Nearest Neighbor |
| RF | Random Forest |
| TF | Term Frequency |
| TF-IDF | Term Frequency Inverse Document Frequency |

| SVD | Singular Value Decomposition |
| LSA | Latent Sematic Analysis |
| BGD | Batch Gradient Descent |
| CM | Confusion Matrix |
| ROC | Receiver Operating Characteristics |
| AUC | Area Under Curve |
| TP | True Positive |
| TN | True Negative |
| FP | False Positive |
| FN | False Negative |

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
