# Peer review of "Identification of Malignancies from Free-Text Histopathology Reports Using a Multi-Model Supervised Machine Learning Approach"

_information, doi:10.3390/info11090455_

Round 1

Reviewer 1 Report

This paper approaches what is recently a very popular in NLP, that is, cáncer reports.

The paper follows a straightforward structure. A few basic supervised classifiers are trained with minimal preprocessing.

The paper is reasonably well-written and presented. However, there are a number of issues that prevent me from recommending to accept this work in its present state, which I detail in the following.

In the preprocessing, they plotted a word cloud on the result text to determine the word representation before data cleaning of the result text and they generated character count, word count and unique word count before data cleaning. After pre-processing they excluded rows where content was completely lost due to pre-processing, could you tell me what rows were excluded?

The supervised approach is very basic. Besides some limited report-specific preprocessing, there is neither feature engineering nor automatic feature learning, e.g., with a deep model. 

You evaluated model performance by running non-optimized classification algorithms. What parameters are considered non-optimized for each algorithms? Why did you use Gaussian Naive Bayes and not used Multinomial Naive Bayes? Why haven't you use the lineal/logistic regression classifier? Could you make a comparison between the non-optimized and optimized parameters for each classifier? There are no obvious reasons to choose the four classifiers chosen

Since you do not use previous work with comparable results in your work, it would have been better to use all possible optimized classifiers to do their work. There are no obvious reasons to choose the four classifiers chosen.

No error analysis or qualitative study is presented. What kind of report are more difficult to classify and why?

No introspection is presented, despite the models being quite transparent. What are the most informative words, bigrams and trigrams?

Table 4 is not explained, what data does it refer to? what does "No/Other labels" column mean? how is the classification of table 4 done? There is a big difference between the SGD and RF classification of the "No/Other label” in Malignant predicted (6,408 versus 12,451), why is this? Is it a good idea to choose SGD as the primary model? What is the meaning of primary model for you?

Author Response

This paper approaches what is recently a very popular in NLP, that is, cáncer reports.

The paper follows a straightforward structure. A few basic supervised classifiers are trained with minimal preprocessing.

The paper is reasonably well-written and presented. However, there are a number of issues that prevent me from recommending to accept this work in its present state, which I detail in the following.

In the preprocessing, they plotted a word cloud on the result text to determine the word representation before data cleaning of the result text and they generated character count, word count and unique word count before data cleaning. After pre-processing they excluded rows where content was completely lost due to pre-processing, could you tell me what rows were excluded?

15 rows returned null values after pre-processing, before pre-processing these rows had, tabs and new lines that had not actual words. Refer to page 3 line 6 to 7, second paragraph of pre-processing sub-heading

The supervised approach is very basic. Besides some limited report-specific preprocessing, there is neither feature engineering nor automatic feature learning, e.g., with a deep model. 

We performed feature engineering, as this allowed us to convert the raw text into classification features using TF-IDF vectorizer, we then review the most important words according to TF-IDF vectorizer but were not contributing to the overall classification. Lastly we reduced the classification features using SVD to the best 100 features. Page 3 line 8 to 25 sub heading feature engineering

You evaluated model performance by running non-optimized classification algorithms. What parameters are considered non-optimized for each algorithms? Why did you use Gaussian Naive Bayes and not used Multinomial Naive Bayes? Why haven't you use the lineal/logistic regression classifier? Could you make a comparison between the non-optimized and optimized parameters for each classifier? There are no obvious reasons to choose the four classifiers chosen

We optimized all the algorithms in the model, we also added logistic regression classifier, and we described each algorithm briefly in the manuscript to help the reader understand how each algorithm functions. However we did not include Multinomial Naïve Bayes, this is because it generated a ValueError: Negative values in data passed to MultinomialNB (input X), since SVD generate features with negative values, MNB does not accept negative values that features will generate. Pages 4 to 5. Also refer to page 17 line 1 to 3

Since you do not use previous work with comparable results in your work, it would have been better to use all possible optimized classifiers to do their work. There are no obvious reasons to choose the four classifiers chosen.

We included all classifiers, optimized and get the classification results from each. Page 10 to 13

No error analysis or qualitative study is presented. What kind of report are more difficult to classify and why?

We performed error analysis by calculating error rates from each classification output using data that had labels. Table 6 page 15 line 1.

No introspection is presented, despite the models being quite transparent. What are the most informative words, bigrams and trigrams?

We attached a sample of TF-IDF most informative words in the supplementary section Table 8, Page 20.

Table 4 is not explained, what data does it refer to? what does "No/Other labels" column mean? how is the classification of table 4 done? There is a big difference between the SGD and RF classification of the "No/Other label” in Malignant predicted (6,408 versus 12,451), why is this? Is it a good idea to choose SGD as the primary model? What is the meaning of primary model for you?

Table 5 is a cross tabulation of the classification output, label before are the labels that were assigned to the report by the pathologist or the laboratory technologists. In case where there was no label assigned we label such reports as “No label”. We considered SGD as a primary object due to its performance during training and optimization, however this actually changed with the actual classification. Our objective was to explore various classifiers and report on their performance in performing classification of histopathology reports. Page 14

Reviewer 2 Report

  1. Delete or shorten the subsection of 2.11. Evaluation.
  2. Prepare Figure 3 and Figure 4 in more compact way.
  3. Deeper experimental analysis is required.
  4. Conclusion part is too simple to accept.

Author Response

  1. Delete or shorten the subsection of 2.11. Evaluation.
    • We dropped all the formulas of the evaluation as these made the subsection to be every long. Page 6, line 10 to 18
  2. Prepare Figure 3 and Figure 4 in more compact way.
    • Figures presentations changed, we also added more models in the final classification, which translated to more figure for the Confusing Matrices and Receiver Operating Characteristics. Pages 10 to 13
  3. Deeper experimental analysis is required.
    • We performed error analysis Table 6 page 15 line 1, and introduced feature engineering techniques page 5 last paragraph.
  4. Conclusion part is too simple to accept.
    • We expanded in the conclusion by recommending the classifiers that can be used in the classification of the histopathology reports. Page 16, line 10 to 17

Round 2

Reviewer 1 Report

The authors have now answered my comments correctly and have improved the points requested. They have done a comprehensive study on ML classifiers so I think it could be accepted for publication.

Author Response

No comment raise by the reviewer.